# GRADIENT DESCENT ALIGNS THE LAYERS OF DEEP LINEAR NETWORKS

**Ziwei Ji & Matus Telgarsky**
Department of Computer Science
University of Illinois at Urbana-Champaign
{ziweiji2,mjt}@illinois.edu

## ABSTRACT

This paper establishes risk convergence and asymptotic weight matrix alignment — a form of implicit regularization — of gradient flow and gradient descent when applied to deep linear networks on linearly separable data. In more detail, for gradient flow applied to strictly decreasing loss functions (with similar results for gradient descent with particular decreasing step sizes): (i) the risk converges to 0; (ii) the normalized $i^{\text{th}}$ weight matrix asymptotically equals its rank-1 approximation $u_i v_i^{\top}$; (iii) these rank-1 matrices are aligned across layers, meaning $|v_{i+1}^{\top} u_i| \to 1$. In the case of the logistic loss (binary cross entropy), more can be said: the linear function induced by the network — the product of its weight matrices — converges to the same direction as the maximum margin solution. This last property was identified in prior work, but only under assumptions on gradient descent which here are implied by the alignment phenomenon.

## 1 INTRODUCTION

Efforts to explain the effectiveness of gradient descent in deep learning have uncovered an exciting possibility: it not only finds solutions with low error, but also biases the search for low complexity solutions which generalize well (Zhang et al., 2017; Bartlett et al., 2017; Soudry et al., 2017; Gunasekar et al., 2018).

This paper analyzes the implicit regularization of gradient descent and gradient flow on deep linear networks and linearly separable data. For strictly decreasing losses, the optimum is at infinity, and we establish various *alignment phenomena*:

- For each weight matrix $W_i$, the corresponding normalized weight matrix $W_i/\|W_i\|_F$ asymptotically equals its rank-1 approximation $u_i v_i^{\top}$, where the Frobenius norm $\|W_i\|_F$ satisfies $\|W_i\|_F \to \infty$. In other words, $\|W_i\|_2/\|W_i\|_F \to 1$, and asymptotically only the rank-1 approximation of $W_i$ contributes to the final predictor, a form of implicit regularization.

- Adjacent rank-1 weight matrix approximations are aligned: $|v_{i+1}^{\top} u_i| \to 1$.

- For the logistic loss, the first right singular vector $v_1$ of $W_1$ is aligned with the data, meaning $v_1$ converges to the unique maximum margin predictor $\bar{u}$ defined by the data. Moreover, the linear predictor induced by the network, $w_{\text{prod}} := W_L \cdots W_1$, is also aligned with the data, meaning $w_{\text{prod}}/\|w_{\text{prod}}\| \to \bar{u}$.

Simultaneously, this work proves that the risk is globally optimized: it asymptotes to 0. Alignment and risk convergence are proved simultaneously; the phenomena are coupled within the proofs.

Since the layers align, they can be viewed as a *minimum norm solution*: they do not "waste norm" on components which are killed off when the layers are multiplied together. Said another way, given data $((x_i, y_i))_{i=1}^n$, the normalized matrices $(W_1/\|W_1\|_F, \ldots, W_L/\|W_L\|_F)$ asymptotically solve a maximum margin problem which demands *all* weight matrices be small, not merely their product:

$$\max_{\substack{W_L \in \mathbb{R}^{1 \times d_{L-1}} \\ \|W_L\|_F = 1}} \cdots \max_{\substack{W_1 \in \mathbb{R}^{d_1 \times d_0} \\ \|W_1\|_F = 1}} \min_i y_i (W_L \cdots W_1) x_i.$$

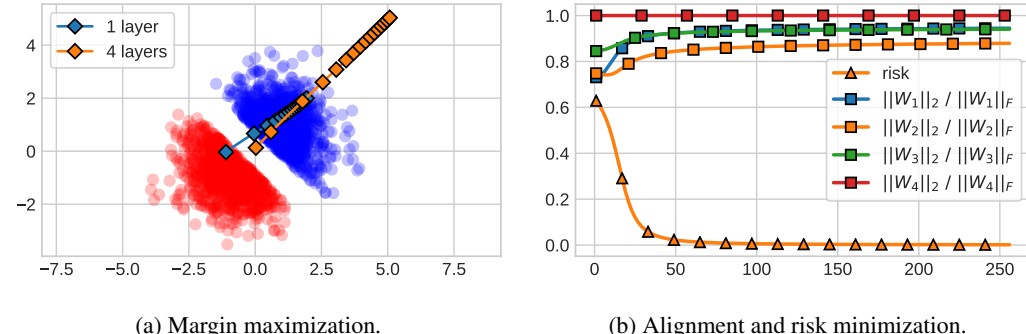

(a) Margin maximization.  (b) Alignment and risk minimization.

Figure 1: Visualization of margin maximization and self-regularization of layers on synthetic data with a 4-layer linear network compared to a 1-layer network (a linear predictor). Figure 1a shows the convergence of 1-layer and 4-layer networks to the same margin-maximizing linear predictor on positive (blue) and negative (red) separable data. Figure 1b shows the convergence of $\|W_i\|_2/\|W_i\|_F$ to 1 on each layer, plotted against the risk.

The paper is organized as follows. This introduction continues with related work, notation, and assumptions in Sections 1.1 and 1.2. The analysis of gradient flow is in Section 2, and gradient descent is analyzed in Section 3. The paper closes with future directions in Section 4; a particular highlight is a preliminary experiment on CIFAR-10 which establishes empirically that a form of the alignment phenomenon occurs on the standard nonlinear network AlexNet.

## 1.1 RELATED WORK

On the implicit regularization of gradient descent, Soudry et al. (2017) show that for linear predictors and linearly separable data, the gradient descent iterates converge to the same direction as the maximum margin solution. Ji & Telgarsky (2018) further characterize such an implicit bias for general nonseparable data. Gunasekar et al. (2018) consider gradient descent on fully connected linear networks and linear convolutional networks. In particular, for the exponential loss, assuming the risk is minimized to 0 and the gradients converge in direction, they show that the whole network converges in direction to the maximum margin solution. These two assumptions are on the gradient descent process itself, and specifically the second one might be hard to interpret and justify. Compared with Gunasekar et al. (2018), this paper *proves* that the risk converges to 0 and the weight matrices align; moreover the proof here proves the properties simultaneously, rather than assuming one and deriving the other. Lastly, Arora et al. (2018) show for deep linear networks (and later Du et al. (2018) for ReLU networks) that gradient flow does not change the difference between squared Frobenius norms of any two layers. We use a few of these tools in our proofs; please see Sections 2 and 3 for details.

For a smooth (nonconvex) function, Lee et al. (2016) show that any strict saddle can be avoided almost surely with small step sizes. If there are only countably many saddle points and they are all strict, and if gradient descent iterates converge, then this implies (almost surely) they converge to a local minimum. In the present work, since there is no finite local minimum, gradient descent will go to infinity and never converge, and thus these results of Lee et al. (2016) do not show that the risk converges to 0.

There has been a rich literature on linear networks. Saxe et al. (2013) analyze the learning dynamics of deep linear networks, showing that they exhibit some learning patterns similar to nonlinear networks, such as a long plateau followed by a rapid risk drop. Arora et al. (2018) show that depth can help accelerate optimization. On the landscape properties of deep linear networks, Lu & Kawaguchi (2017); Laurent & von Brecht (2017) show that under various structural assumptions, all local optima are global. Zhou & Liang (2018) give a necessary and sufficient characterization of critical points for deep linear networks.

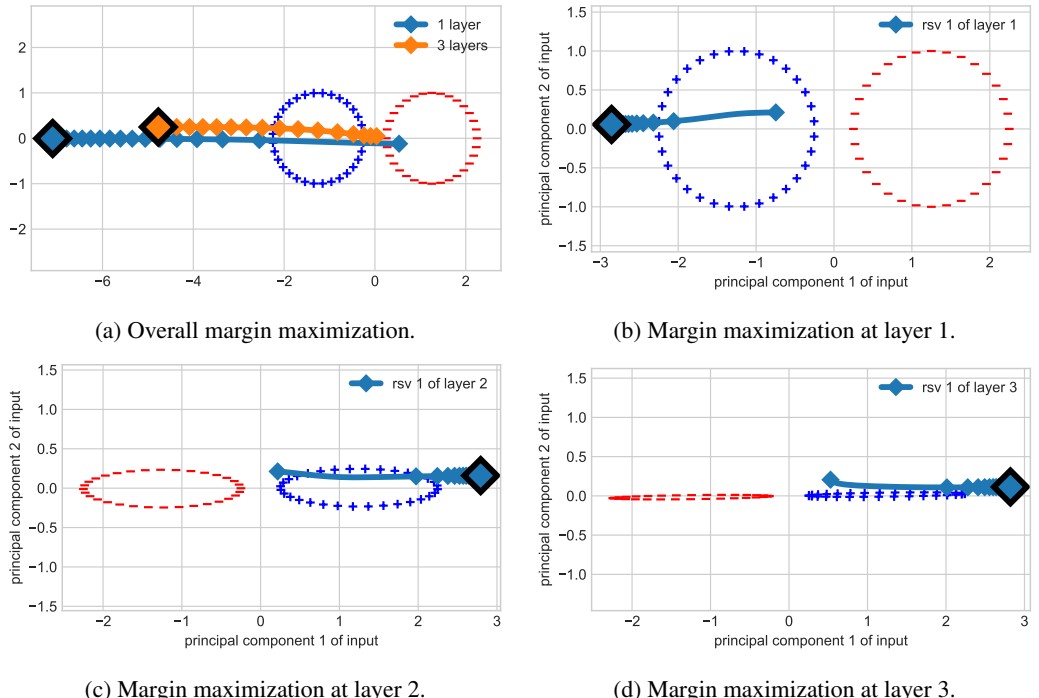

(a) Overall margin maximization.

(b) Margin maximization at layer 1.

(c) Margin maximization at layer 2.

(d) Margin maximization at layer 3.

Figure 2: A visualization of inter-layer alignment on data consisting of two well-separated circles with a 3-layer linear network. Figure 2a depicts, as in Figure 1a, that optimizing 1- and 3-layer linear networks finds the same maximum margin solution. The other three plots show the data as it is mapped through progressively more and more layers. Due to alignment, the product $W_i \cdots W_1$ becomes $u_i \bar{u}^\top$, where $u_i$ is the top left singular vector of $W_i$, which means that asymptotically the mapped data will be well separated and lie along the span of $u_i$, as depicted by the flattening in Figures 2b to 2d. Additionally, these three subfigures show that the top right singular vector $v_{i+1}$ of the subsequent layer is aligned with this $u_i$, which in these plots (with principal component axes) corresponds to following a horizontal line.

## 1.2 NOTATION, SETTING, AND ASSUMPTIONS

Consider a data set $\{(x_i, y_i)\}_{i=1}^n$, where $x_i \in \mathbb{R}^d$, $\|x_i\| \leq 1$, and $y_i \in \{-1, +1\}$. The data set is assumed to be linearly separable, i.e., there exists a unit vector $u$ which correctly classifies every data point: for any $1 \leq i \leq n$, $y_i \langle u, x_i \rangle > 0$. Furthermore, let $\gamma := \max_{\|u\|=1} \min_{1 \leq i \leq n} y_i \langle u, x_i \rangle > 0$ denote the maximum margin, and $\bar{u} := \arg\max_{\|u\|=1} \min_{1 \leq i \leq n} y_i \langle u, x_i \rangle$ denote the maximum margin solution (the solution to the hard-margin SVM).

A linear network of depth $L$ is parameterized by weight matrices $W_L, \ldots, W_1$, where $W_k \in \mathbb{R}^{d_k \times d_{k-1}}$, $d_0 = d$, and $d_L = 1$. Let $W = (W_L, \ldots, W_1)$ denote all parameters of the network. The (empirical) risk induced by the network is given by

$$\mathcal{R}(W) = \mathcal{R}(W_L, \ldots, W_1) = \frac{1}{n}\sum_{i=1}^n \ell(y_i W_L \cdots W_1 x_i) = \frac{1}{n}\sum_{i=1}^n \ell(\langle w_{\text{prod}}, z_i \rangle),$$

where $w_{\text{prod}} := (W_L \cdots W_1)^\top$, and $z_i := y_i x_i$.

The loss $\ell$ is assumed to be continuously differentiable, unbounded, and strictly decreasing to 0. Examples include the exponential loss $\ell_{\exp}(x) = e^{-x}$ and the logistic loss $\ell_{\log}(x) = \ln(1 + e^{-x})$.

**Assumption 1.** $\ell' < 0$ is continuous, $\lim_{x \to -\infty} \ell(x) = \infty$ and $\lim_{x \to \infty} \ell(x) = 0$.

This paper considers gradient flow and gradient descent, where gradient flow $\{W(t) | t \geq 0, t \in \mathbb{R}\}$ can be interpreted as gradient descent with infinitesimal step sizes. It starts from some $W(0)$ at

$t = 0$, and proceeds as

$$\frac{\mathrm{d}W(t)}{\mathrm{d}t} = -\nabla \mathcal{R}\left(W(t)\right).$$

By contrast, gradient descent $\left\{W(t)\middle|t \geq 0, t \in \mathbb{Z}\right\}$ is a discrete-time process given by

$$W(t+1) = W(t) - \eta_t \nabla \mathcal{R}\left(W(t)\right),$$

where $\eta_t$ is the step size at time $t$.

We assume that the initialization of the network is not a critical point and induces a risk no larger than the risk of the trivial linear predictor $0$.

**Assumption 2.** The initialization $W(0)$ satisfies $\nabla \mathcal{R}\left(W(0)\right) \neq 0$ and $\mathcal{R}\left(W(0)\right) \leq \mathcal{R}(0) = \ell(0)$.

It is natural to require that the initialization is not a critical point, since otherwise gradient flow/descent will never make a progress. The requirement $\mathcal{R}\left(W(0)\right) \leq \mathcal{R}(0)$ can be easily satisfied, for example, by making $W_1(0) = 0$ and $W_L(0) \cdots W_2(0) \neq 0$. On the other hand, if $\mathcal{R}\left(W(0)\right) > \mathcal{R}(0)$, gradient flow/descent may never minimize the risk to $0$. Proofs of those claims are given in Appendix A.

## 2 RESULTS FOR GRADIENT FLOW

In this section, we consider gradient flow. Although impractical when compared with gradient descent, gradient flow can simplify the analysis and highlight proof ideas. For convenience, we usually use $W$, $W_k$, and $w_{\mathrm{prod}}$, but they all change with (the continuous time) $t$. Only proof sketches are given here; detailed proofs are deferred to Appendix B.

### 2.1 RISK CONVERGENCE

One key property of gradient flow is that it never increases the risk:

$$\frac{\mathrm{d}\mathcal{R}(W)}{\mathrm{d}t} = \left\langle \nabla \mathcal{R}(W), \frac{\mathrm{d}W}{\mathrm{d}t} \right\rangle = -\|\nabla \mathcal{R}(W)\|^2 = -\sum_{k=1}^{L} \left\| \frac{\partial \mathcal{R}}{\partial W_k} \right\|_F^2 \leq 0. \tag{1}$$

We now state the main result: under Assumptions 1 and 2, gradient flow minimizes the risk, $W_k$ and $w_{\mathrm{prod}}$ all go to infinity, and the alignment phenomenon occurs.

**Theorem 1.** *Under Assumptions 1 and 2, gradient flow iterates satisfy the following properties:*

- $\lim_{t \to \infty} \mathcal{R}(W) = 0.$

- *For any $1 \leq k \leq L$, $\lim_{t \to \infty} \|W_k\|_F = \infty$.*

- *For any $1 \leq k \leq L$, letting $(u_k, v_k)$ denote the first left and right singular vectors of $W_k$,*

$$\lim_{t \to \infty} \left\| \frac{W_k}{\|W_k\|_F} - u_k v_k^\top \right\|_F = 0.$$

*Moreover, for any $1 \leq k < L$, $\lim_{t \to \infty} \left|\langle v_{k+1}, u_k \rangle\right| = 1$. As a result,*

$$\lim_{t \to \infty} \left| \left\langle \frac{w_{\mathrm{prod}}}{\prod_{k=1}^{L} \|W_k\|_F}, v_1 \right\rangle \right| = 1,$$

*and thus $\lim_{t \to \infty} \|w_{\mathrm{prod}}\| = \infty$.*

Theorem 1 is proved using two lemmas, which may be of independent interest. To show the ideas, let us first introduce a little more notation. Recall that $\mathcal{R}(W)$ denotes the empirical risk induced by the deep linear network $W$. Abusing the notation a little, for any linear predictor $w \in \mathbb{R}^d$, we also use $\mathcal{R}(w)$ to denote the risk induced by $w$. With this notation, $\mathcal{R}(W) = \mathcal{R}(w_{\mathrm{prod}})$, while

$$\nabla \mathcal{R}(w_{\mathrm{prod}}) = \frac{1}{n} \sum_{i=1}^{n} \ell'\left(\langle w_{\mathrm{prod}}, z_i \rangle\right) z_i = \frac{1}{n} \sum_{i=1}^{n} \ell'\left(W_L \cdots W_1 z_i\right) z_i$$

is in $\mathbb{R}^d$ and different from $\nabla \mathcal{R}(W)$, which has $\sum_{k=1}^{L} d_k d_{k-1}$ entries, as given below:

$$\frac{\partial \mathcal{R}}{\partial W_k} = W_{k+1}^\top \cdots W_L^\top \nabla \mathcal{R}(w_{\mathrm{prod}})^\top W_1^\top \cdots W_{k-1}^\top.$$

Furthermore, for any $R > 0$, let

$$B(R) = \left\{ W \Big| \max_{1 \leq k \leq L} \|W_k\|_F \leq R \right\}.$$

The first lemma shows that for any $R > 0$, the time spent by gradient flow in $B(R)$ is finite.

**Lemma 1.** *Under Assumption 1 and 2, for any $R > 0$, there exists a constant $\epsilon(R) > 0$, such that for any $t \geq 1$ and any $W \in B(R)$, $\|\partial \mathcal{R}/\partial W_1\|_F \geq \epsilon(R)$. As a result, gradient flow spends a finite amount of time in $B(R)$ for any $R > 0$, and $\max_{1 \leq k \leq L} \|W_k\|_F$ is unbounded.*

Here is the proof sketch. If $\|W_k\|_F$ are bounded, then $\|\nabla \mathcal{R}(w_{\mathrm{prod}})\|$ will be lower bounded by a positive constant, therefore if $\|\partial \mathcal{R}/\partial W_1\|_F = \|W_L \cdots W_2\| \|\nabla \mathcal{R}(w_{\mathrm{prod}})\|$ can be arbitrarily small, then $\|W_L \cdots W_2\|$ and $\|w_{\mathrm{prod}}\|$ can also be arbitrarily small, and thus $\mathcal{R}(W)$ can be arbitrarily close to $\mathcal{R}(0)$. This cannot happen after $t = 1$, otherwise it will contradict Assumption 2 and eq. (1).

To proceed, we need the following properties of linear networks from prior work (Arora et al., 2018; Du et al., 2018). For any time $t \geq 0$ and any $1 \leq k < L$,

$$W_{k+1}^\top(t)W_{k+1}(t) - W_{k+1}^\top(0)W_{k+1}(0) = W_k(t)W_k^\top(t) - W_k(0)W_k^\top(0). \tag{2}$$

To see this, just notice that

$$W_{k+1}^\top \frac{\partial \mathcal{R}}{\partial W_{k+1}} = W_{k+1}^\top \cdots W_L^\top \nabla \mathcal{R}(w_{\mathrm{prod}})^\top W_1^\top \cdots W_k^\top = \frac{\partial \mathcal{R}}{\partial W_k} W_k^\top.$$

Taking the trace on both sides of eq. (2), we have

$$\left\|W_{k+1}(t)\right\|_F^2 - \left\|W_{k+1}(0)\right\|_F^2 = \left\|W_k(t)\right\|_F^2 - \left\|W_k(0)\right\|_F^2. \tag{3}$$

In other words, the difference between the squares of Frobenius norms of any two layers remains a constant. Together with Lemma 1, it implies that all $\|W_k\|_F$ are unbounded.

However, even if $\|W_k\|_F$ are large, it does not follow necessarily that $\|w_{\mathrm{prod}}\|$ is also large. Lemma 2 shows that this is indeed true: for gradient flow, as $\|W_k\|_F$ get larger, adjacent layers also get more aligned to each other, which ensures that their product also has a large norm.

For $1 \leq k \leq L$, let $\sigma_k$, $u_k$, and $v_k$ denote the first singular value (the 2-norm), the first left singular vector, and the first right singular vector of $W_k$, respectively. Furthermore, define

$$D := \left( \max_{1 \leq k \leq L} \|W_k(0)\|_F^2 \right) - \|W_L(0)\|_F^2 + \sum_{k=1}^{L-1} \left\| W_k(0)W_k^\top(0) - W_{k+1}^\top(0)W_{k+1}(0) \right\|_2,$$

which depends only on the initialization. If for any $1 \leq k < L$, $W_k(0)W_k^\top(0) = W_{k+1}^\top(0)W_{k+1}(0)$, then $D = 0$.

**Lemma 2.** *The gradient flow iterates satisfy the following properties:*

- *For any $1 \leq k \leq L$, $\|W_k\|_F^2 - \|W_k\|_2^2 \leq D$.*

- *For any $1 \leq k < L$, $\langle v_{k+1}, u_k \rangle^2 \geq 1 - \left(D + \|W_{k+1}(0)\|_2^2 + \|W_k(0)\|_2^2\right) / \|W_{k+1}\|_2^2$.*

- *Suppose $\max_{1 \leq k \leq L} \|W_k\|_F \to \infty$, then $\left| \left\langle w_{\mathrm{prod}}/\prod_{k=1}^{L} \|W_k\|_F, v_1 \right\rangle \right| \to 1$.*

The proof is based on eq. (2) and eq. (3). If $W_k(0)W_k^\top(0) = W_{k+1}^\top(0)W_{k+1}(0)$, then eq. (2) gives that $W_{k+1}$ and $W_k$ have the same singular values, and $W_{k+1}$'s right singular vectors and $W_k$'s left singular vectors are the same. If it is true for any two adjacent layers, since $W_L$ is a row vector, all layers have rank 1. With general initialization, we have similar results when $\|W_k\|_F$ is large enough so that the initialization is negligible. Careful calculations give the exact results in Lemma 2.

An interesting point is that the implicit regularization result in Lemma 2 helps establish risk convergence in Theorem 1. Specifically, by Lemma 2, if all layers have large norms, $\|W_L \cdots W_2\|$ will be large. If the risk is not minimized to 0, $\|\nabla \mathcal{R}(w_{\mathrm{prod}})\|$ will be lower bounded by a positive constant, and thus $\|\partial \mathcal{R}/\partial W_1\|_F = \|W_L \cdots W_2\| \|\nabla \mathcal{R}(w_{\mathrm{prod}})\|$ will be large. Invoking eq. (1), Lemma 1 and eq. (3) gives a contradiction. Since the risk has no finite optimum, $\|W_k\|_F \to \infty$.

## 2.2 Convergence to the maximum margin solution

Here we focus on the exponential loss $\ell_{\exp}(x) = e^{-x}$ and the logistic loss $\ell_{\log}(x) = \ln(1 + e^{-x})$. In addition to risk convergence, these two losses also enable gradient descent to find the maximum margin solution.

To get such a strong convergence, we need one more assumption on the data set. Recall that $\gamma = \max_{\|u\|=1} \min_{1 \le i \le n} \langle u, z_i \rangle > 0$ denotes the maximum margin, and $\bar{u}$ denotes the unique maximum margin predictor which attains this margin $\gamma$. Those data points $z_i$ for which $\langle \bar{u}, z_i \rangle = \gamma$ are called support vectors.

**Assumption 3.** The support vectors span the whole space $\mathbb{R}^d$.

Assumption 3 appears in prior work Soudry et al. (2017), and can be satisfied in many cases: for example, it is almost surely true if the number of support vectors is larger than or equal to $d$ and the data set is sampled from some density w.r.t. the Lebesgue measure. It can also be relaxed to the situation that the support vectors and the whole data set span the same space; in this case $\nabla \mathcal{R}(w_{\mathrm{prod}})$ will never leave this space, and we can always restrict our attention to this space.

With Assumption 3, we can state the main theorem.

**Theorem 2.** *Under Assumptions 2 and 3, for almost all data and for losses $\ell_{\exp}$ and $\ell_{\log}$, then $\lim_{t \to \infty} |\langle v_1, \bar{u} \rangle| = 1$, where $v_1$ is the first right singular vector of $W_1$. As a result, $\lim_{t \to \infty} w_{\mathrm{prod}} / \prod_{k=1}^{L} \|W_k\|_F = \bar{u}$.*

Before summarizing the proof, we can simplify both theorems into the following *minimum norm* property mentioned in the introduction.

**Corollary 1.** *Under Assumptions 2 and 3, for almost all data and for losses $\ell_{\exp}$ and $\ell_{\log}$,*

$$\min_i y_i \left( \frac{W_L}{\|W_L\|_F} \cdots \frac{W_1}{\|W_1\|_F} \right) x_i \quad \xrightarrow[t \to \infty]{} \quad \max_{\substack{A_L \in \mathbb{R}^{1 \times d_{L-1}} \\ \|A_L\|_F = 1}} \cdots \max_{\substack{A_1 \in \mathbb{R}^{d_1 \times d_0} \\ \|A_1\|_F = 1}} \min_i y_i (A_L \cdots A_1) x_i.$$

Theorem 2 relies on two structural lemmas. The first one is based on a similar almost-all argument due to Soudry et al. (2017, Lemma 12). Let $S \subset \{1, \ldots, n\}$ denote the set of indices of support vectors.

**Lemma 3.** *Under Assumption 3, if the data set is sampled from some density w.r.t. the Lebesgue measure, then with probability 1,*

$$\alpha := \min_{|\xi|=1, \xi \perp \bar{u}} \max_{i \in S} \langle \xi, z_i \rangle > 0.$$

Let $\bar{u}^\perp$ denote the orthogonal complement of $\mathrm{span}(\bar{u})$, and let $\Pi_\perp$ denote the projection onto $\bar{u}^\perp$. We prove that if $\|\Pi_\perp w\|$ is large enough, gradient flow starting from $w$ will tend to decrease $\|\Pi_\perp w\|$.

**Lemma 4.** *Under Assumption 3, for almost all data, $\ell_{\exp}$ and $\ell_{\log}$, and any $w \in \mathbb{R}^d$, if $\langle w, \bar{u} \rangle \ge 0$ and $\|\Pi_\perp w\|$ is larger than $^{1+\ln(n)}/_\alpha$ for $\ell_{\exp}$ or $^{2n}/_{e\alpha}$ for $\ell_{\log}$, then $\langle \Pi_\perp w, \nabla \mathcal{R}(w) \rangle \ge 0$.*

With Lemma 3 and Lemma 4 in hand, we can prove Theorem 2. Let $\Pi_\perp W_1$ denote the projection of rows of $W_1$ onto $\bar{u}^\perp$. Notice that

$$\Pi_\perp w_{\mathrm{prod}} = \left( W_L \ldots W_2 (\Pi_\perp W_1) \right)^\top \quad \text{and} \quad \frac{\mathrm{d}\|\Pi_\perp W_1\|_F^2}{\mathrm{d}t} = -2 \langle \Pi_\perp w_{\mathrm{prod}}, \nabla \mathcal{R}(w_{\mathrm{prod}}) \rangle.$$

If $\|\Pi_\perp W_1\|_F$ is large compared with $\|W_1\|_F$, since layers become aligned, $\|\Pi_\perp w_{\mathrm{prod}}\|$ will also be large, and then Lemma 4 implies that $\|\Pi_\perp W_1\|_F$ will not increase. At the same time, $\|W_1\|_F \to \infty$, and thus for large enough $t$, $\|\Pi_\perp W_1\|_F$ must be very small compared with $\|W_1\|_F$. Many details need to be handled to make this intuition exact; the proof is given in Appendix B.

## 3 Results for gradient descent

One key property of gradient flow which is used in the previous proofs is that it never increases the risk, which is not necessarily true for gradient descent. However, for smooth losses (i.e, with

Lipschitz continuous derivatives), we can design some decaying step sizes, with which gradient descent never increases the risk, and basically the same results hold as in the gradient flow case. Deferred proofs are given in Appendix C.

We make the following additional assumption on the loss, which is satisfied by the logistic loss $\ell_{\log}$.

**Assumption 4.** $\ell'$ is $\beta$-Lipschitz (i.e, $\ell$ is $\beta$-smooth), and $|\ell'| \leq G$ (i.e., $\ell$ is $G$-Lipschitz).

Under Assumption 4, the risk is also a smooth function of $W$, if all layers are bounded.

**Lemma 5.** *Suppose $\ell$ is $\beta$-smooth. If $R \geq 1$, then $\beta(R) = 2L^2 R^{2L-2}(\beta + G)$, and $\mathcal{R}(W)$ is a $\beta(R)$-smooth function on the set $B(R) = \{W \mid \|W_k\|_F \leq R, 1 \leq k \leq L\}$.*

Smoothness ensures that for any $W, V \in B(R)$, $\mathcal{R}(W) - \mathcal{R}(V) \leq \langle \nabla \mathcal{R}(V), W - V \rangle + \beta(R)\|W-V\|^2/2$ (see Bubeck et al. (2015) Lemma 3.4). In particular, if we choose some $R$ and set a constant step size $\eta_t = 1/\beta(R)$, then as long as $W(t+1)$ and $W(t)$ are both in $B(R)$,

$$\mathcal{R}\big(W(t+1)\big) - \mathcal{R}\big(W(t)\big) \leq \left\langle \nabla\mathcal{R}\big(W(t)\big), -\eta_t \nabla\mathcal{R}\big(W(t)\big) \right\rangle + \frac{\beta(R)\eta_t^2}{2}\left\|\nabla\mathcal{R}\big(W(t)\big)\right\|^2$$

$$= -\frac{1}{2\beta(R)}\left\|\nabla\mathcal{R}\big(W(t)\big)\right\|^2 = -\frac{\eta_t}{2}\left\|\nabla\mathcal{R}\big(W(t)\big)\right\|^2. \tag{4}$$

In other words, the risk does not increase at this step. However, similar to gradient flow, the gradient descent iterate will eventually escape $B(R)$, which may increase the risk.

**Lemma 6.** *Under Assumption 1, 2 and 4, suppose gradient descent is run with a constant step size $1/\beta(R)$. Then there exists a time $t$ when $W(t) \notin B(R)$, in other words, $\max_{1 \leq k \leq L} \|W_k(t)\|_F > R$.*

Fortunately, this issue can be handled by adaptively increasing $R$ and correspondingly decreasing the step sizes, formalized in the following assumption.

**Assumption 5.** The step size $\eta_t = \min\{1/\beta(R_t), 1\}$, where $R_t$ satisfies $W(t) \in B(R_t - 1)$, and if $W(t+1) \in B(R_t - 1)$, $R_{t+1} = R_t$.

Assumption 5 can be satisfied by a line search, which ensures that the gradient descent update is not too aggressive and the boundary $R$ is increased properly.

With the additional Assumptions 4 and 5, exactly the same theorems can be proved for gradient descent. We restate them briefly here.

**Theorem 3.** *Under Assumption 1, 2, 4, and 5, gradient descent satisfies*

- $\lim_{t\to\infty} \mathcal{R}\big(W(t)\big) = 0$.

- *For any $1 \leq k \leq L$, $\lim_{t\to\infty} \|W_k(t)\|_F = \infty$.*

- $\lim_{t\to\infty}\left|\left\langle w_{\mathrm{prod}}(t)/\prod_{k=1}^L \|W_k(t)\|_F, v_1(t) \right\rangle\right| = 1$, *where $v_1(t)$ is the first right singular vector of $W_1(t)$.*

**Theorem 4.** *Under Assumptions 2, 3 and 5, for the logistic loss $\ell_{\log}$ and almost all data, $\lim_{t\to\infty}\left|\langle v_1(t), \bar{u}\rangle\right| = 1$, and $\lim_{t\to\infty} w_{\mathrm{prod}}(t)/\prod_{k=1}^L \|W_k(t)\|_F = \bar{u}$.*

**Corollary 2.** *Under Assumptions 2, 3 and 5, for the logistic loss $\ell_{\log}$ and almost all data,*

$$\min_i y_i \left( \frac{W_L}{\|W_L\|_F} \cdots \frac{W_1}{\|W_1\|_F} \right) x_i \quad \xrightarrow[t\to\infty]{} \quad \max_{\substack{A_L \in \mathbb{R}^{1 \times d_{L-1}} \\ \|A_L\|_F = 1}} \cdots \max_{\substack{A_1 \in \mathbb{R}^{d_1 \times d_0} \\ \|A_1\|_F = 1}} \min_i y_i (A_L \cdots A_1) x_i.$$

Proofs of Theorem 3 and 4 are given in Appendix C, and are basically the same as the gradient flow proofs. The key difference is that an error of $\sum_{t=0}^{\infty} \eta_t^2 \|\nabla\mathcal{R}(W(t))\|^2$ will be introduced in many parts of the proof. However, it is bounded in light of eq. (4):

$$\sum_{t=0}^{\infty} \eta_t^2 \left\|\nabla\mathcal{R}\big(W(t)\big)\right\|^2 \leq \sum_{t=0}^{\infty} \eta_t \left\|\nabla\mathcal{R}\big(W(t)\big)\right\|^2 \leq 2\mathcal{R}\big(W(0)\big).$$

Since all weight matrices go to infinity, such a bounded error does not matter asymptotically, and thus proofs still go through.

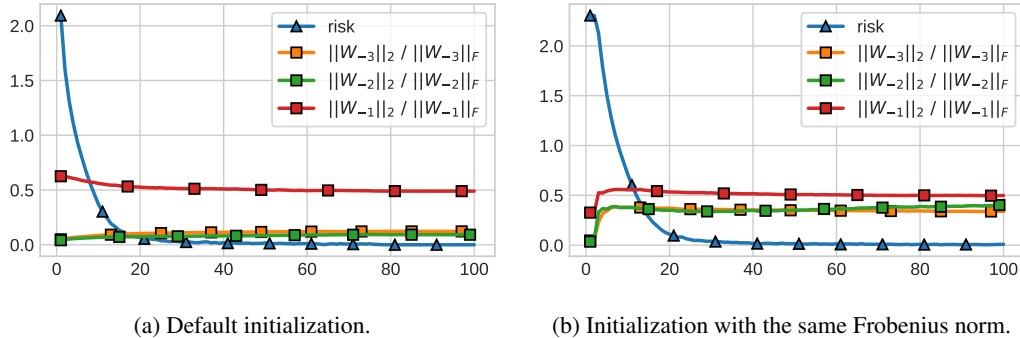

(a) Default initialization.

(b) Initialization with the same Frobenius norm.

Figure 3: Risk and alignment of dense layers (the ratio $\|W_i\|_2/\|W_i\|_F$) of (nonlinear!) AlexNet on CIFAR-10. Figure 3a uses default PyTorch initialization, while Figure 3b forces initial Frobenius norms to be equal amongst dense layers.

## 4 SUMMARY AND FUTURE DIRECTIONS

This paper rigorously proves that, for deep linear networks on linearly separable data, gradient flow and gradient descent minimize the risk to $0$, align adjacent weight matrices, and align the first right singular vector of the first layer to the maximum margin solution determined by the data. There are many potential future directions; a few are as follows.

**Convergence rates and practical step sizes.** This paper only proves asymptotic convergence, moreover for adaptive step sizes depending on the current weight matrix norms. A refined analysis with rates for practical step sizes (e.g., constant step sizes) would allow the algorithm to be compared to other methods which also globally optimize this objective, would suggest ways to improve step sizes and initialization, and ideally even exhibit a sensitivity to the network architecture and suggest how it could be improved.

**Nonseparable data and nonlinear networks.** Real-world data is generally not linearly separable, but nonlinear deep networks can reliably decrease the risk to $0$, even with random labels (Zhang et al., 2017). This seems to suggest that a nonlinear notion of separability is at play; is there some way to adapt the present analysis?

The present analysis is crucially tied to the alignment of weight matrices: alignment and risk are analyzed simultaneously. Motivated by this, consider a preliminary experiment, presented in Figure 3, where stochastic gradient descent was used to minimize the risk of a standard AlexNet on CIFAR-10 (Krizhevsky et al., 2012; Krizhevsky & Hinton, 2009).

Even though there are ReLUs, max-pooling layers, and convolutional layers, the alignment phenomenon is occurring in a reduced form on the dense layers (the last three layers of the network). Specifically, despite these weight matrices having shape $(1024, 4096)$, $(4096, 4096)$, and $(4096, 10)$ the key alignment ratios $\|W_i\|_2/\|W_i\|_F$ are much larger than their respective lower bounds $(1024^{-1/2}, 4096^{-1/2}, 10^{-1/2})$. Two initializations were tried: default PyTorch initialization, and a Gaussian initialization forcing all initial Frobenius norms to be just $4$, which is suggested by the norm preservation property in the analysis and removes noise in the weights.

ACKNOWLEDGEMENTS

The authors are grateful for support from the NSF under grant IIS-1750051. This grant allowed them to focus on research, and when combined with an NVIDIA GPU grant, led to the creation of their beloved GPU machine DUTCHCRUNCH.

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

## A    REGARDING ASSUMPTION 2

Suppose $W_1(0) = 0$ while $W_L(0) \cdots W_2(0) \neq 0$. First of all, $W_L(0) \cdots W_1(0) = 0$ and thus $\mathcal{R}\left(W(0)\right) = \mathcal{R}(0)$. Moreover,

$$\left\langle \nabla \mathcal{R}\left(w_{\text{prod}}(0)\right), \bar{u} \right\rangle = \frac{1}{n} \sum_{i=1}^{n} \ell'(0)\langle z_i, \bar{u}\rangle \leq \ell'(0)\gamma < 0,$$

which implies $\nabla \mathcal{R}\left(w_{\text{prod}}(0)\right) \neq 0$ and $\partial \mathcal{R}/\partial W_1 = \left(W_L(0) \cdots W_2(0)\right)^{\top} \nabla \mathcal{R}\left(w_{\text{prod}}(0)\right)^{\top} \neq 0$.

On the other hand, if $\mathcal{R}\left(W(0)\right) > \mathcal{R}(0)$, gradient flow/descent may never minimize the risk to 0. For example, suppose the network has two layers, and both the input and output have dimension 1; the network just computes the dot product of two vectors $w_1$ and $w_2$. Consider minimizing $\mathcal{R}(w_1, w_2) = \exp\left(-\langle w_1, w_2\rangle\right)$. If $w_1(0) = -w_2(0) \neq 0$, then $\mathcal{R}\left(w_1(0), w_2(0)\right) = \exp\left(\|w_1\|^2\right) > \exp(0)$. It is easy to verify that for any $t$, $w_1(t) = -w_2(t)$, and $\mathcal{R}\left(w_1(t), w_2(t)\right) \geq \exp(0) > 0$.

## B    OMITTED PROOFS FROM SECTION 2

*Proof of Lemma 1.* Fix an arbitrary $R > 0$. If the claim is not true, then for any $\epsilon > 0$, there exists some $t \geq 1$ such that $\|W_k\|_F \leq R$ for all $k$ while $\left\|\partial \mathcal{R}/\partial W_1\right\|_F^2 \leq \epsilon^2$, which means

$$\left\|\frac{\partial \mathcal{R}}{\partial W_1}\right\|_F^2 = \left\|W_2^{\top} \cdots W_L^{\top} \nabla \mathcal{R}(w_{\text{prod}})^{\top}\right\|_F^2 = \|W_L \cdots W_2\|^2 \|\nabla \mathcal{R}(w_{\text{prod}})\|^2 \leq \epsilon^2.$$

Since $\|w_{\text{prod}}\| \leq R^L$, we have

$$\langle \nabla \mathcal{R}(w_{\text{prod}}), \bar{u}\rangle = \frac{1}{n} \sum_{i=1}^{n} \ell'\left(\langle w_{\text{prod}}, z_i\rangle\right) \langle z_i, \bar{u}\rangle \leq \frac{1}{n} \sum_{i=1}^{n} \ell'\left(\langle w_{\text{prod}}, z_i\rangle\right) \gamma \leq -M\gamma,$$

where $-M = \max_{-R^L \leq x \leq R^L} \ell'(x)$. Since $\ell'$ is continuous and the domain is bounded, the maximum is attained and negative, and thus $M > 0$. Therefore $\|\nabla \mathcal{R}(w_{\text{prod}})\| \geq M\gamma$, and thus $\|W_L \cdots W_2\| \leq \epsilon/M\gamma$. Since $\|W_1\|_F \leq R$, we further have $\|w_{\text{prod}}\| \leq \epsilon R/M\gamma$. In other words, after $t = 1$, $\|w_{\text{prod}}\|$ may be arbitrarily small, which implies $\mathcal{R}\left(w_{\text{prod}}\right)$ can be arbitrarily close to $\mathcal{R}(0)$.

On the other hand, by Assumption 2, $d\mathcal{R}(W)/dt = -\|\nabla \mathcal{R}(W)\|^2 < 0$ at $t = 0$. This implies that $\mathcal{R}\left(W(1)\right) < \mathcal{R}\left(W(0)\right)$, and for any $t \geq 1$, $\mathcal{R}\left(W(t)\right) \leq \mathcal{R}\left(W(1)\right) < \mathcal{R}\left(W(0)\right) \leq \mathcal{R}(0)$, which is a contradiction.

Since the risk is always positive, we have

$$\mathcal{R}\left(W(0)\right) \geq \int_{t=0}^{\infty} \sum_{k=1}^{L} \left\|\frac{\partial \mathcal{R}}{\partial W_k}\right\|_F^2 dt$$

$$\geq \int_{t=0}^{\infty} \left\|\frac{\partial \mathcal{R}}{\partial W_1}\right\|_F^2 dt$$

$$\geq \int_{t=0}^{\infty} \left\|\frac{\partial \mathcal{R}}{\partial W_1}\right\|_F^2 \mathbb{1}\left[\max_{1 \leq k \leq L} \|W_k\|_F \leq R\right] dt$$

$$\geq \int_{t=1}^{\infty} \left\|\frac{\partial \mathcal{R}}{\partial W_1}\right\|_F^2 \mathbb{1}\left[\max_{1 \leq k \leq L} \|W_k\|_F \leq R\right] dt$$

$$\geq \epsilon(R)^2 \int_{t=1}^{\infty} \mathbb{1}\left[\max_{1 \leq k \leq L} \|W_k\|_F \leq R\right] dt,$$

which implies gradient flow only spends a finite amount of time in $\{W | \max_{1 \leq k \leq L} \|W_k\|_F \leq R\}$. This directly implies that $\max_{1 \leq k \leq L} \|W_k\|_F$ is unbounded. $\square$

*Proof of Lemma 2.* The first claim is true for $k = L$ since $W_L$ is a row vector. For any $1 \le k < L$, recall that Arora et al. (2018); Du et al. (2018) give the following relation:

$$W_{k+1}^\top(t)W_{k+1}(t) - W_{k+1}^\top(0)W_{k+1}(0) = W_k(t)W_k^\top(t) - W_k(0)W_k^\top(0). \tag{5}$$

Let $A_{k,k+1} = W_k(0)W_k^\top(0) - W_{k+1}^\top(0)W_{k+1}(0)$. By eq. (5) and the definition of singular vectors and singular values, we have

$$
\begin{aligned}
\sigma_k^2 &\ge v_{k+1}^\top W_k W_k^\top v_{k+1} \\
&= v_{k+1}^\top W_{k+1}^\top W_{k+1} v_{k+1} + v_{k+1}^\top A_{k,k+1} v_{k+1} \\
&= \sigma_{k+1}^2 + v_{k+1}^\top A_{k,k+1} v_{k+1} \\
&\ge \sigma_{k+1}^2 - \|A_{k,k+1}\|_2.
\end{aligned}
\tag{6}
$$

Moreover, by taking the trace on both sides of eq. (5), we have

$$
\begin{aligned}
\|W_k\|_F^2 = \mathrm{tr}\left(W_k W_k^\top\right) &= \mathrm{tr}\left(W_{k+1}^\top W_{k+1}\right) + \mathrm{tr}\left(W_k(0)W_k^\top(0)\right) - \mathrm{tr}\left(W_{k+1}^\top(0)W_{k+1}(0)\right) \\
&= \|W_{k+1}\|_F^2 + \|W_k(0)\|_F^2 - \|W_{k+1}(0)\|_F^2.
\end{aligned}
\tag{7}
$$

Summing eq. (6) and eq. (7) from $k$ to $L - 1$, we get

$$\|W_k\|_F^2 - \|W_k\|_2^2 \le \|W_k(0)\|_F^2 - \|W_L(0)\|_F^2 + \sum_{k'=k}^{L-1} \|A_{k',k'+1}\|_2 \le D. \tag{8}$$

Next we prove that singular vectors get aligned. Consider $u_k^\top W_{k+1}^\top W_{k+1} u_k$. On one hand, similar to eq. (6), we can get that

$$
\begin{aligned}
u_k^\top W_{k+1}^\top W_{k+1} u_k &= u_k^\top W_k W_k^\top u_k - u_k^\top W_k(0)W_k^\top(0)u_k + u_k^\top W_{k+1}^\top(0)W_{k+1}(0)u_k \\
&\ge u_k^\top W_k W_k^\top u_k - u_k^\top W_k(0)W_k^\top(0)u_k \\
&\ge \sigma_k^2 - \|W_k(0)\|_2^2.
\end{aligned}
\tag{9}
$$

On the other hand, it follows from the definition of singular vectors and eq. (8) that

$$
\begin{aligned}
u_k^\top W_{k+1}^\top W_{k+1} u_k &= \langle u_k, v_{k+1}\rangle^2 \sigma_{k+1}^2 + u_k^\top \left(W_{k+1}^\top W_{k+1} - v_{k+1}\sigma_{k+1}^2 v_{k+1}^\top\right) u_k \\
&\le \langle u_k, v_{k+1}\rangle^2 \sigma_{k+1}^2 + \|W_{k+1}\|_F^2 - \|W_{k+1}\|_2^2 \\
&\le \langle u_k, v_{k+1}\rangle^2 \sigma_{k+1}^2 + D.
\end{aligned}
\tag{10}
$$

Combining eq. (9) and eq. (10), we get

$$\sigma_k^2 \le \langle u_k, v_{k+1}\rangle^2 \sigma_{k+1}^2 + D + \|W_k(0)\|_2^2. \tag{11}$$

Similar to eq. (9), we can get

$$\sigma_k^2 \ge v_{k+1}^\top W_k W_k^\top v_{k+1} \ge \sigma_{k+1}^2 - \|W_{k+1}(0)\|_2^2.$$

Therefore

$$\frac{\sigma_k^2}{\sigma_{k+1}^2} \ge 1 - \frac{\|W_{k+1}(0)\|_2^2}{\sigma_{k+1}^2}. \tag{12}$$

Combining eq. (11) and eq. (12), we finally get

$$\langle u_k, v_{k+1}\rangle^2 \ge 1 - \frac{D + \|W_k(0)\|_2^2 + \|W_{k+1}(0)\|_2^2}{\sigma_{k+1}^2}.$$

Regarding the last claim, first recall that since the difference between the squares of Frobenius norms of any two layers is a constant, $\max_{1 \le k \le L} \|W_k\|_F \to \infty$ implies $\|W_k\|_F \to \infty$ for any $k$. We further have the following.

- Since $\|W_k\|_F^2 - \|W_k\|_2^2 \le D$, $\|W_k\|_2 \to \infty$ for any $k$, and $W_k/\|W_k\|_F \to u_k v_k^\top$.

- Since $\|W_k\|_2 \to \infty$, $|\langle u_k, v_{k+1}\rangle| \to 1$.

As a result,

$$
\left|\left\langle \frac{w_{\text{prod}}}{\prod_{k=1}^{L}\|W_k\|_F}, v_1 \right\rangle\right| = \left|\left\langle \prod_{k=1}^{L} \frac{W_k}{\|W_k\|_F}, v_1 \right\rangle\right|
$$

$$
\to \left|\left\langle \prod_{k=1}^{L} u_i v_i^\top, v_1 \right\rangle\right|
$$

$$
\to 1.
$$

$\square$

*Proof of Theorem 1.* Suppose for some $\epsilon > 0$, $\mathcal{R}(W) \geq \epsilon$ for any $t$. Then there exists some $1 \leq j \leq n$ such that $\ell\left(\langle w_{\text{prod}}, z_j\rangle\right) \geq \epsilon$, and thus $\langle w_{\text{prod}}, z_j\rangle \leq \ell^{-1}(\epsilon)$. On the other hand, since $\mathcal{R}(W) \leq \mathcal{R}(0) = \ell(0)$, $\ell\left(\langle w_{\text{prod}}, z_j\rangle\right) \leq n\ell(0)$, and thus $\langle w_{\text{prod}}, z_j\rangle \geq \ell^{-1}\left(n\ell(0)\right)$. Let $-M = \max_{\ell^{-1}\left(n\ell(0)\right) \leq x \leq \ell^{-1}(\epsilon/n)} \ell'(x) < 0$, we have for any $t$,

$$
\langle \nabla \mathcal{R}(w_{\text{prod}}), \bar{u}\rangle = \frac{1}{n}\sum_{i=1}^{n} \ell'\left(\langle w_{\text{prod}}, z_i\rangle\right)\langle z_i, \bar{u}\rangle
$$

$$
\leq \frac{1}{n}\sum_{i=1}^{n} \ell'\left(\langle w_{\text{prod}}, z_i\rangle\right)\gamma
$$

$$
\leq \frac{1}{n}\ell'\left(\langle w_{\text{prod}}, z_j\rangle\right)\gamma
$$

$$
\leq \frac{-M\gamma}{n} < 0,
$$

and thus $\|\nabla \mathcal{R}(w_{\text{prod}})\| \geq M\gamma/n$.

Similar to the proof of Lemma 2, we can show that if $\|W_k\|_F \to \infty$,

$$
\left|\left\langle \frac{(W_L \cdots W_2)^\top}{\|W_k\|_F \cdots \|W_2\|_F}, v_2 \right\rangle\right| \to 1.
$$

In other words, there exists some $C > 0$, such that when $\min_{1 \leq k \leq L}\|W_k\|_F > C$, $\|W_L \cdots W_2\| \geq \|W_k\|_F \cdots \|W_2\|_F/2 > C^L/2$.

Lemma 1 shows that gradient flow spends a finite amount of time in $\left\{W \big| \max_{1 \leq k \leq L}\|W_k\|_F \leq R\right\}$ for any $R > 0$. Since the difference between the squares of Frobenius norms of any two layers is a constant, gradient flow also spends a finite amount of time in $\left\{W \big| \min_{1 \leq k \leq L}\|W_k\|_F \leq C\right\}$. Now we have

$$
\mathcal{R}(W(0)) \geq \int_{t=0}^{\infty} \sum_{k=1}^{L} \left\|\frac{\partial \mathcal{R}}{\partial W_k}\right\|_F^2 \, \mathrm{d}t
$$

$$
\geq \int_{t=0}^{\infty} \left\|\frac{\partial \mathcal{R}}{\partial W_1}\right\|_F^2 \, \mathrm{d}t
$$

$$
= \int_{t=0}^{\infty} \|W_L \cdots W_2\|^2 \|\nabla \mathcal{R}(w_{\text{prod}})\|^2 \, \mathrm{d}t
$$

$$
\geq \int_{t=0}^{\infty} \|W_L \cdots W_2\|^2 \|\nabla \mathcal{R}(w_{\text{prod}})\|^2 \mathbb{1}\left[W \Big| \min_{1 \leq k \leq L}\|W_k\|_F > C\right] \mathrm{d}t
$$

$$
\geq \left(\frac{M\gamma}{n}\right)^2 \left(\frac{C^L}{2}\right)^2 \int_{t=0}^{\infty} \mathbb{1}\left[W \Big| \min_{1 \leq k \leq L}\|W_k\|_F > C\right] \mathrm{d}t
$$

$$
= \infty,
$$

which is a contradiction. Therefore $\mathcal{R}(\epsilon) \to 0$. This further implies $\|W_k\|_F \to \infty$, since $\mathcal{R}(W)$ has no finite optimum. Finally, invoking Lemma 2 proves the final claim of Theorem 1. $\qquad\square$

*Proof of Lemma 3.* Soudry et al. (2017) Lemma 12 proves that, with probability 1, there are at most $d$ support vectors, and moreover, the $i$-th support vector $z_i$ has a positive dual variable $\alpha_i$, such that $\sum_{i \in S} \alpha_i z_i = \bar{u}$.

Suppose there exists some $\xi \perp \bar{u}$, such that $\max_{i \in S}\langle \xi, z_i \rangle \le 0$. Since

$$\sum_{i \in S} \alpha_i \langle \xi, z_i \rangle = \left\langle \xi, \sum_{i \in S} \alpha_i z_i \right\rangle = \langle \xi, \bar{u} \rangle = 0,$$

we actually have $\langle \xi, z_i \rangle = 0$ for all $i \in S$. This is impossible under Assumption 3, since the support vectors span the whole space. $\qquad\square$

*Proof of Lemma 4.* For the sake of presentation, we leave out the subscript in $z_i$ and denote a data point by $z$ generally. For any data point $z$ and predictor $w$, let $z_\perp$ and $w_\perp$ denote their projection onto $\bar{u}^\perp$. Let $z' \in \arg\max_{i \in S}\langle -w_\perp, z \rangle$, and thus $\langle -w_\perp, z' \rangle \ge \alpha\|w_\perp\|$.

For $\ell_{\exp}$, we have

$$\langle w_\perp, \nabla \mathcal{R}(w) \rangle = \sum_z \frac{1}{n} \left[ -\exp\left(-\langle w, z \rangle\right) \right] \langle w_\perp, z_\perp \rangle$$

$$= \sum_z \frac{1}{n} \left[ \exp\left(\langle -w, z \rangle\right) \right] \langle -w_\perp, z_\perp \rangle$$

$$\ge \frac{1}{n} \exp\left(\langle -w, z' \rangle\right) \langle -w_\perp, z'_\perp \rangle + \sum_{\langle z_\perp, w_\perp \rangle \ge 0} \frac{1}{n} \exp\left(\langle -w, z \rangle\right) \langle -w_\perp, z_\perp \rangle. \quad (13)$$

The first part can be lower bounded as below (recall that $\langle -w_\perp, z'_\perp \rangle = \langle -w_\perp, z' \rangle \ge \alpha\|w_\perp\|$)

$$\frac{1}{n} \exp\left(\langle -w, z' \rangle\right) \langle -w_\perp, z'_\perp \rangle = \frac{1}{n} \exp\left(\langle -w, \gamma\bar{u} \rangle\right) \exp\left(\langle -w_\perp, z'_\perp \rangle\right) \langle -w_\perp, z'_\perp \rangle$$

$$\ge \frac{1}{n} \exp\left(-\langle w, \gamma\bar{u} \rangle\right) \exp\left(\alpha\|w_\perp\|\right) \alpha\|w_\perp\|. \quad (14)$$

To bound the second part, first notice that since we assume $\langle w, \bar{u} \rangle \ge 0$, for any $z$,

$$\langle w, z - \gamma\bar{u} \rangle = \langle w, z_\perp \rangle + \langle w, z - \gamma\bar{u} - z_\perp \rangle \ge \langle w, z_\perp \rangle = \langle w_\perp, z_\perp \rangle. \quad (15)$$

The reason is that every data point has margin at least $\gamma$, and thus $z - \gamma\bar{u} - z_\perp = c\bar{u}$ for some $c \ge 0$. Using eq. (15), we can bound the second part of eq. (13).

$$\sum_{\langle z_\perp, w_\perp \rangle \ge 0} \frac{1}{n} \exp\left(\langle -w, z \rangle\right) \langle -w_\perp, z_\perp \rangle$$

$$= \sum_{\langle z_\perp, w_\perp \rangle \ge 0} \frac{1}{n} \exp\left(\langle -w, \gamma\bar{u} \rangle\right) \exp\left(\langle -w, z - \gamma\bar{u} \rangle\right) \langle -w_\perp, z_\perp \rangle$$

$$\ge \sum_{\langle z_\perp, w_\perp \rangle \ge 0} \frac{1}{n} \exp\left(\langle -w, \gamma\bar{u} \rangle\right) \exp\left(\langle -w_\perp, z_\perp \rangle\right) \langle -w_\perp, z_\perp \rangle$$

$$\ge \sum_{\langle z_\perp, w_\perp \rangle \ge 0} \frac{1}{n} \exp\left(\langle -w, \gamma\bar{u} \rangle\right) \left(-\frac{1}{e}\right)$$

$$\ge \quad \exp\left(\langle -w, \gamma\bar{u} \rangle\right) \left(-\frac{1}{e}\right). \quad (16)$$

On the third line eq. (15) is applied. The fourth line applies the property that $f(x) = -xe^{-x} \ge -1/e$ when $x \ge 0$.

Combining eq. (13), eq. (14) and eq. (16), we get

$$\langle w_\perp, \nabla \mathcal{R}(w) \rangle \geq \exp\left(\langle -w, \gamma \bar{u} \rangle\right) \left(\frac{1}{n} \exp\left(\alpha \|w_\perp\|\right) \alpha \|w_\perp\| - \frac{1}{e}\right).$$

As long as $\|w_\perp\| \geq (1 + \ln(n))/\alpha$, $\langle w_\perp, \nabla \mathcal{R}(w) \rangle \geq 0$.

For $\ell_{\log}$, similar to eq. (13), we have

$$
\begin{aligned}
\langle w_\perp, \nabla \mathcal{R}(w) \rangle &\geq \frac{1}{n} \frac{\exp\left(\langle -w, z' \rangle\right)}{1 + \exp\left(\langle -w, z' \rangle\right)} \langle -w_\perp, z'_\perp \rangle + \sum_{\langle z_\perp, w_\perp \rangle \geq 0} \frac{1}{n} \frac{\exp\left(\langle -w, z \rangle\right)}{1 + \exp\left(\langle -w, z \rangle\right)} \langle -w_\perp, z_\perp \rangle \\
&\geq \frac{1}{n} \frac{\exp\left(\langle -w, z' \rangle\right)}{1 + \exp\left(\langle -w, z' \rangle\right)} \langle -w_\perp, z'_\perp \rangle + \sum_{\langle z_\perp, w_\perp \rangle \geq 0} \frac{1}{n} \exp\left(\langle -w, z \rangle\right) \langle -w_\perp, z_\perp \rangle.
\end{aligned}
$$

(17)

The second part of eq. (17) can be bounded again by eq. (16). To bound the first part of eq. (17), first notice that (recall $\langle w, \bar{u} \rangle \geq 0$)

$$\exp\left(\langle -w, z' \rangle\right) = \exp\left(\langle -w, \gamma \bar{u} \rangle\right) \exp\left(\langle -w_\perp, z'_\perp \rangle\right) \leq \exp\left(\langle -w_\perp, z'_\perp \rangle\right). \tag{18}$$

Using eq. (18), and recall that $\langle -w_\perp, z'_\perp \rangle = \langle -w_\perp, z' \rangle \geq \alpha \|w_\perp\| \geq 0$, we can bound the first part of eq. (17) as below.

$$
\begin{aligned}
\frac{1}{n} \frac{\exp\left(\langle -w, z' \rangle\right)}{1 + \exp\left(\langle -w, z' \rangle\right)} \langle -w_\perp, z'_\perp \rangle &= \frac{1}{n} \exp\left(\langle -w, \gamma \bar{u} \rangle\right) \frac{\exp\left(\langle -w_\perp, z'_\perp \rangle\right)}{1 + \exp\left(\langle -w, z' \rangle\right)} \langle -w_\perp, z'_\perp \rangle \\
&\geq \frac{1}{n} \exp\left(\langle -w, \gamma \bar{u} \rangle\right) \frac{\exp\left(\langle -w_\perp, z'_\perp \rangle\right)}{1 + \exp\left(\langle -w_\perp, z'_\perp \rangle\right)} \langle -w_\perp, z'_\perp \rangle \\
&\geq \frac{1}{2n} \exp\left(\langle -w, \gamma \bar{u} \rangle\right) \langle -w_\perp, z'_\perp \rangle \\
&\geq \frac{1}{2n} \exp\left(\langle -w, \gamma \bar{u} \rangle\right) \alpha \|w_\perp\|.
\end{aligned}
$$

(19)

Combining eq. (17), eq. (19) and eq. (16), we get

$$\langle w_\perp, \nabla \mathcal{R}(w) \rangle \geq \exp\left(\langle -w, \gamma \bar{u} \rangle\right) \left(\frac{1}{2n} \alpha \|w_\perp\| - \frac{1}{e}\right).$$

As long as $\|w_\perp\| \geq 2n/e\alpha$, $\langle w_\perp, \nabla \mathcal{R}(w) \rangle \geq 0$. $\qquad \square$

*Proof of Theorem 2.* Recall that

$$\frac{\mathrm{d}W_1}{\mathrm{d}t} = -\frac{\partial \mathcal{R}}{\partial W_1} = -W_2^\top \cdots W_L^\top \nabla \mathcal{R}(w_{\mathrm{prod}})^\top,$$

and thus

$$\frac{\mathrm{d}\|W_1\|_F^2}{\mathrm{d}t} = \left\langle W_1, \frac{\mathrm{d}W_1}{\mathrm{d}t} \right\rangle = -2 \langle w_{\mathrm{prod}}, \nabla \mathcal{R}(w_{\mathrm{prod}}) \rangle. \tag{20}$$

Let $\Pi_{\bar{u}}$ denote the projection onto $\mathrm{span}(\bar{u})$, and let $\Pi_\perp$ denote the projection onto $\bar{u}^\perp$. Also let $\Pi_{\bar{u}} W_1$ and $\Pi_\perp W_1$ denote the projection of rows of $W_1$ onto $\mathrm{span}(\bar{u})$ and $\bar{u}^\perp$, respectively. Notice that

$$\Pi_{\bar{u}} w_{\mathrm{prod}} = \left(W_L \cdots W_2 (\Pi_{\bar{u}} W_1)\right)^\top, \quad \text{and} \quad \Pi_\perp w_{\mathrm{prod}} = \left(W_L \cdots W_2 (\Pi_\perp W_1)\right)^\top.$$

We further have

$$\frac{\mathrm{d}\|\Pi_\perp W_1\|_F^2}{\mathrm{d}t} = -2 \langle \Pi_\perp w_{\mathrm{prod}}, \nabla \mathcal{R}(w_{\mathrm{prod}}) \rangle. \tag{21}$$

Let $W_1 = u_1 \sigma_1 v_1^\top + S$. We have $\|S\|_2 \leq \sigma_{1,2} \leq \sqrt{\sigma_{1,2}^2} \leq \sqrt{\|W_1\|_F^2 - \|W_1\|_2^2} \leq \sqrt{D}$, where $\sigma_{1,2}$ is the second singular value of $W_1$ and $D$ is the constant introduced in Lemma 2. Then

$$\Pi_\perp W_1 = u_1 \sigma_1 \left(\Pi_\perp v_1\right)^\top + \Pi_\perp S,$$

and

$$\|\Pi_\perp W_1\|_F \le \left\|u_1\sigma_1\left(\Pi_\perp v_1\right)^\top\right\|_F + \|\Pi_\perp S\|_F = \sigma_1\|\Pi_\perp v_1\| + \|\Pi_\perp S\|_F \le \sigma_1\|\Pi_\perp v_1\| + \sqrt{dD}.$$

It follows that

$$\|\Pi_\perp v_1\| \ge \frac{\|\Pi_\perp W_1\|_F}{\sigma_1} - \frac{\sqrt{dD}}{\sigma_1} \ge \frac{\|\Pi_\perp W_1\|_F}{\|W_1\|_F} - \frac{\sqrt{dD}}{\|W_1\|_2}. \tag{22}$$

Fix an arbitrary $\epsilon > 0$. By Theorem 1, we can find some $t_0$ large enough such that for any $t \ge t_0$:

1. $\sqrt{dD}/\|W_1\|_2 \le \epsilon/3$.

2. $\left\|w_{\mathrm{prod}}/\|W_L\|_F\cdots\|W_1\|_F - v_1\right\| \le \epsilon/3$, or $\left\|w_{\mathrm{prod}}/\|W_L\|_F\cdots\|W_1\|_F + v_1\right\| \le \epsilon/3$.

3. $\|W_L\|_F\cdots\|W_1\|_F \ge 3K/\epsilon$, where $K$ is the threshold given in Lemma 4, i.e., $1+\ln(n)/\alpha$ for $\ell_{\mathrm{exp}}$, $2n/e\alpha$ for $\ell_{\mathrm{log}}$.

4. $\mathcal{R}(W) \le \ell(0)/n$, which implies $\langle w_{\mathrm{prod}}, z_i\rangle \ge 0$ for all $1 \le i \le n$. By Lemma 3, there always exists a support vector $z$ for which $\langle \Pi_\perp w_{\mathrm{prod}}, z\rangle \le 0$, and therefore $\langle w_{\mathrm{prod}}, \bar{u}\rangle \ge 0$.

Suppose for some $t \ge t_0$, $\|\Pi_\perp W_1\|_F/\|W_1\|_F \ge \epsilon$. By eq. (22) and bullet 1 above, $\|\Pi_\perp v_1\| \ge 2\epsilon/3$. Bullet 2 above then gives $\left\|\Pi_\perp w_{\mathrm{prod}}/\|W_L\|_F\cdots\|W_1\|_F\right\| \ge \epsilon/3$, which together with bullet 3 above implies $\|\Pi_\perp w_{\mathrm{prod}}\| \ge K$. Since also $\langle w_{\mathrm{prod}}, \bar{u}\rangle \ge 0$, we can apply Lemma 4 and get that $\langle \Pi_\perp w_{\mathrm{prod}}, \nabla \mathcal{R}(w_{\mathrm{prod}})\rangle \ge 0$. In light of eq. (21), $\mathrm{d}\|\Pi_\perp W_1\|_F^2/\mathrm{d}t \le 0$.

On the other hand, since after $t \ge t_0$, $\langle w_{\mathrm{prod}}, z_i\rangle \ge 0$, we have $\mathrm{d}\|W_1\|_F^2/\mathrm{d}t \ge 0$ by eq. (20). Therefore $\|\Pi_\perp W_1\|_F/\|W_1\|_F$ will not increase, and since $\|W_1\|_F \to \infty$, it will eventually drop below $\epsilon$, and will never exceed $\epsilon$ again. Therefore,

$$\limsup_{t\to\infty} \frac{\|\Pi_\perp W_1\|_F}{\|W_1\|_F} \le \epsilon.$$

Since $\epsilon$ is arbitrary, we have

$$\limsup_{t\to\infty} \frac{\|\Pi_\perp W_1\|_F}{\|W_1\|_F} = 0,$$

and thus $\lim_{t\to\infty}\left|\langle v_1, \bar{u}\rangle\right| = 1$. An application of Theorem 1 gives the other part of Theorem 2. $\square$

*Proof of Corollary 1.* By Theorem 2,

$$\min_i y_i \left(\frac{W_L}{\|W_L\|_F}\cdots\frac{W_1}{\|W_1\|_F}\right)x_i \quad\longrightarrow\quad \min_i y_i\bar{u}^\top x_i.$$

Next, pick arbitrary unit vectors $a_i \in \mathbb{R}^{d_i}$, and note

$$\max_{\substack{A_1\in\mathbb{R}^{d_1\times d_0}\\ \|A_1\|_F=1}} \cdots \max_{\substack{A_L\in\mathbb{R}^{1\times d_{L-1}}\\ \|A_L\|_F=1}} \min_i y_i\left(A_L\cdots A_1\right)x_i \ge \min_i y_i\left((1a_{L-1}^\top)(a_{L-1}a_{L-2}^\top)\cdots(a_2a_1^\top)(a_1\bar{u}^\top)\right)x_i$$

$$= \min_i y_i\bar{u}^\top x_i.$$

On the other hand, if matrices $(A_L, \ldots, A_1)$ are feasible, then

$$\|A_L\cdots A_1\|_2 \le \|A_L\|_2\cdots\|A_2\|_2\|A_1\|_F \le \|A_L\|_F\cdots\|A_1\|_F \le 1,$$

whereby

$$\min_i y_i(A_L\cdots A_1)x_i \le \|A_L\cdots A_1\|_2 \min_i y_i\left(\frac{A_L\cdots A_1}{\|A_L\cdots A_1\|_2}\right)x_i$$

$$\le 1\cdot \max_{\|w\|_2=1}\min_i y_i w^\top x_i$$

$$= \min_i y_i\bar{u}^\top x_i.$$

$\square$

## C  OMITTED PROOFS FROM SECTION 3

*Proof of Lemma 5.* Given $W, V \in B(R)$, we need to show that $\|\nabla\mathcal{R}(W) - \nabla\mathcal{R}(V)\| \leq \beta(R)\|W - V\|$ for some $\beta(R)$.

Consider $k = 1$ first. Let $w = (W_L \cdots W_1)^\top$, and $v = (V_L \cdots V_1)^\top$. Since $|\ell'| \leq G$, $\|\nabla\mathcal{R}(w)\|, \|\nabla\mathcal{R}(v)\| \leq G$. We have

$$
\begin{aligned}
\left\| \frac{\partial\mathcal{R}}{\partial W_1} - \frac{\partial\mathcal{R}}{\partial V_1} \right\| &= \left\| W_2^\top \cdots W_L^\top \nabla\mathcal{R}(w) - V_2^\top \cdots V_L^\top \nabla\mathcal{R}(v) \right\| \\
&\leq \left\| W_2^\top \cdots W_L^\top \nabla\mathcal{R}(w) - V_2^\top W_3^\top \cdots W_L^\top \nabla\mathcal{R}(w) \right\| \\
&\quad + \left\| V_2^\top W_3^\top \cdots W_L^\top \nabla\mathcal{R}(w) - V_2^\top \cdots V_L^\top \nabla\mathcal{R}(v) \right\| \\
&\leq R^{L-2} G \|W_2 - V_2\| \\
&\quad + \left\| V_2^\top W_3^\top \cdots W_L^\top \nabla\mathcal{R}(w) - V_2^\top \cdots V_L^\top \nabla\mathcal{R}(v) \right\| \\
&\leq R^{L-2} G \|W - V\| \\
&\quad + \left\| V_2^\top W_3^\top \cdots W_L^\top \nabla\mathcal{R}(w) - V_2^\top \cdots V_L^\top \nabla\mathcal{R}(v) \right\|. \tag{23}
\end{aligned}
$$

Proceeding in this way, we can get

$$
\left\| \frac{\partial\mathcal{R}}{\partial W_1} - \frac{\partial\mathcal{R}}{\partial V_1} \right\| \leq (L-1)R^{L-2}G\|W - V\| + R^{L-1}\|\nabla\mathcal{R}(w) - \nabla\mathcal{R}(v)\|. \tag{24}
$$

Since $\|z_i\| \leq 1$, $\ell'$ is $\beta$-Lipschitz, we have

$$
\|\nabla\mathcal{R}(w) - \nabla\mathcal{R}(v)\| \leq \beta\|w - v\| \leq \beta L R^{L-1}\|W - V\|, \tag{25}
$$

where the last inequality follows from a similar one-by-one replacement procedure as in eq. (23). Combining eq. (24) and eq. (25), we get for $R \geq 1$,

$$
\left\| \frac{\partial\mathcal{R}}{\partial W_1} - \frac{\partial\mathcal{R}}{\partial V_1} \right\| \leq \left( (L-1)R^{L-2}G + \beta L R^{2L-2} \right) \|W - V\| \leq 2LR^{2L-2}(\beta + G)\|W - V\|.
$$

The same procedure can be done for other layers, and together

$$
\|\nabla\mathcal{R}(W) - \nabla\mathcal{R}(V)\| \leq 2L^2 R^{2L-2}(\beta + G)\|W - V\|.
$$

$\square$

*Proof of Lemma 6.* Recall that if $W(t), W(t+1) \in B(R)$ and $\eta_t = 1/\beta(R)$,

$$
\begin{aligned}
\mathcal{R}\left(W(t+1)\right) - \mathcal{R}\left(W(t)\right) &\leq \langle \nabla\mathcal{R}\left(W(t)\right), -\eta_t \nabla\mathcal{R}\left(W(t)\right) \rangle + \frac{\beta(R)\eta_t^2}{2}\left\| \nabla\mathcal{R}\left(W(t)\right) \right\|^2 \\
&= -\frac{1}{2\beta(R)}\left\| \nabla\mathcal{R}\left(W(t)\right) \right\|^2 \\
&= -\frac{\eta_t}{2}\left\| \nabla\mathcal{R}\left(W(t)\right) \right\|^2. \tag{26}
\end{aligned}
$$

Suppose $W(t) \in B(R)$ for all $t$. By Assumption 2 and eq. (26),

$$
\mathcal{R}\left(W(1)\right) \leq \mathcal{R}\left(W(0)\right) - \frac{1}{2\beta(R)}\left\| \nabla\mathcal{R}\left(W(0)\right) \right\|^2 < \mathcal{R}\left(W(0)\right).
$$

By eq. (26), gradient descent never increases the risk, and thus for all $t \geq 1$, $\mathcal{R}\left(W(t)\right) \leq \mathcal{R}\left(W(1)\right) < \mathcal{R}\left(W(0)\right)$. In exactly the same way as in the proof of Lemma 1, one can show that there exists some constant $\epsilon(R) > 0$, so that $\|\partial\mathcal{R}/\partial W_1(t)\|_F \geq \epsilon(R)$ for all $t$. Invoking eq. (26) again, we will get

$$
\mathcal{R}\left(W(0)\right) \geq \sum_{t=0}^{\infty} \frac{1}{2\beta(R)}\epsilon(R)^2 = \infty,
$$

which is a contradiction. Therefore $W(t)$ must go out of $B(R)$ at some time. $\square$

Next we prove Theorem 3 and 4. The proofs depend on several lemmas which are similar to the gradient flow ones. The following Lemma 7 is similar to Lemma 1.

**Lemma 7.** *Under Assumption 1, 2, 4, and 5, gradient descent ensures that*

- $\max_{1 \leq k \leq L} \|W_k(t)\|_F$ *is unbounded.*

- $\sum_{t=0}^{\infty} \eta_t = \infty$.

- *For any $R > 0$, $\sum_{t:W(t) \in B(R)} \eta_t < \infty$.*

*Proof.* By Assumption 5, we always have that $W(t) \in B(R_t)$. Since $\beta(R_t) = 2L^2 R_t^{2L-2}(\beta+G) \geq R_t^{L-1}G$, we have for any $1 \leq k \leq L$,

$$\|W_k(t+1)\|_F \leq \|W_k(t)\|_F + \eta_t \left\|\frac{\partial \mathcal{R}}{\partial W_k(t)}\right\|_F$$
$$\leq \|W_k(t)\|_F + \frac{1}{\beta(R_t)}\left\|\frac{\partial \mathcal{R}}{\partial W_k(t)}\right\|_F$$
$$\leq \|W_k(t)\|_F + \frac{1}{\beta(R_t)}R_t^{L-1}G$$
$$\leq \|W_k(t)\|_F + 1. \tag{27}$$

Moreover, Lemma 6 shows that $R_t \to \infty$. Since $R_{t+1} = R_t$ as long as $W(t+1) \in B(R_t - 1)$, $\max_{1 \leq k \leq L} \|W_k(t)\|_F$ is unbounded.

It then follows that for any $t$, by Cauchy-Schwarz,

$$\left(\sum_{\tau=0}^{t-1} \eta_\tau\right)\left(\sum_{\tau=0}^{t-1} \eta_\tau \left\|\nabla\mathcal{R}\left(W(\tau)\right)\right\|^2\right) \geq \left(\sum_{\tau=0}^{t-1} \eta_\tau \left\|\nabla\mathcal{R}\left(W(\tau)\right)\right\|\right)^2 \to \infty.$$

since by eq. (26),

$$\sum_{\tau=0}^{t-1} \eta_\tau \left\|\nabla\mathcal{R}\left(W(\tau)\right)\right\|^2 \leq 2\mathcal{R}\left(W(0)\right) - 2\mathcal{R}\left(W(t)\right) \leq 2\mathcal{R}\left(W(0)\right),$$

we have $\sum_{t=0}^{\infty} \eta_t = \infty$.

Since under Assumptions 4 and 5 gradient descent never increases the risk, it can be shown in exactly the same as in the proof of Lemma 1 that, for $W(t) \in B(R)$, $\|\partial\mathcal{R}/\partial W_1(t)\|_F \geq \epsilon(R)$ for some constant $\epsilon(R) > 0$. Invoking eq. (26) again, we get that $\sum_{t:W(t) \in B(R)} \eta_t < \infty$. □

The next lemma is an analogy to Lemma 2.

**Lemma 8.** *Under Assumption 1 and 4, the gradient descent iterates satisfy the following properties:*

- *For any $1 \leq k \leq L$, $\|W_k\|_F^2 - \|W_k\|_2^2 \leq D + 2\mathcal{R}\left(W(0)\right)$.*

- *For any $1 \leq k < L$, $\langle v_{k+1}, u_k\rangle^2 \geq 1 - \frac{D+3\mathcal{R}(W(0))+\|W_{k+1}(0)\|_2^2+\|W_k(0)\|_2^2}{\|W_{k+1}\|_2^2}$.*

- *Suppose $\max_{1 \leq k \leq L} \|W_k\|_F \to \infty$, then $\left|\langle w_{\text{prod}}/\prod_{k=1}^{L} \|W_k\|_F, v_1\rangle\right| \to 1$.*

*Proof.* Recall that for any $W$,

$$W_{k+1}^\top \frac{\partial \mathcal{R}}{\partial W_{k+1}} = W_{k+1}^\top \cdots W_L^\top \nabla\mathcal{R}(w_{\text{prod}})^\top W_1^\top \cdots W_k^\top = \frac{\partial \mathcal{R}}{\partial W_k} W_k^\top. \tag{28}$$

For gradient descent iterates, summing eq. (28) from 0 to $t-1$, we get

$$W_{k+1}^\top(t)W_{k+1}(t) - W_{k+1}^\top(0)W_{k+1}(0) + \sum_{\tau=0}^{t-1} \eta_\tau^2 \left(\frac{\partial \mathcal{R}}{\partial W_{k+1}(\tau)}\right)^\top \left(\frac{\partial \mathcal{R}}{\partial W_{k+1}(\tau)}\right)$$
$$= W_k(t)W_k^\top(t) - W_k(0)W_k^\top(0) + \sum_{\tau=0}^{t-1} \eta_\tau^2 \left(\frac{\partial \mathcal{R}}{\partial W_k(\tau)}\right)\left(\frac{\partial \mathcal{R}}{\partial W_k(\tau)}\right)^\top. \tag{29}$$

For any $1 \leq k \leq L$ and any $t$, let

$$P_k(t) = \sum_{\tau=0}^{t-1} \eta_\tau^2 \left( \frac{\partial \mathcal{R}}{\partial W_k(\tau)} \right) \left( \frac{\partial \mathcal{R}}{\partial W_k(\tau)} \right)^\top ,$$

and

$$Q_k(t) = \sum_{\tau=0}^{t-1} \eta_\tau^2 \left( \frac{\partial \mathcal{R}}{\partial W_k(\tau)} \right)^\top \left( \frac{\partial \mathcal{R}}{\partial W_k(\tau)} \right) .$$

We have $\|P_k(t)\|_2 = \|Q_k(t)\|_2 \leq \text{tr}\left(Q_k(t)\right) = \text{tr}\left(P_k(t)\right)$. Moreover, invoking eq. (26),

$$\begin{aligned}
\sum_{k=1}^{L} \text{tr}\left(P_k(t)\right) &= \sum_{k=1}^{L} \sum_{\tau=0}^{t-1} \eta_\tau^2 \left\| \frac{\partial \mathcal{R}}{\partial W_k(\tau)} \right\|_F^2 \\
&= \sum_{\tau=0}^{t-1} \eta_\tau^2 \left\| \nabla \mathcal{R}\left(W(\tau)\right) \right\|^2 \\
&\leq \sum_{\tau=0}^{t-1} \eta_\tau \left\| \nabla \mathcal{R}\left(W(\tau)\right) \right\|^2 \\
&\leq 2\mathcal{R}\left(W(0)\right) - 2\mathcal{R}\left(W(t)\right) \\
&\leq 2\mathcal{R}\left(W(0)\right) .
\end{aligned}$$

(30)

Still let $\sigma_k(t)$, $u_k(t)$ and $v_k(t)$ denote the first singular value, left singular vector and right singular vector of $W_k(t)$. We can then proceed basically in the same way as in the proof of Lemma 2. For example, eq. (6) becomes

$$\sigma_k^2(t) \geq \sigma_{k+1}^2(t) - \|A_{k,k+1}(t)\|_2 - \|P_k(t)\|_2 \geq \sigma_{k+1}^2(t) - \|A_{k,k+1}(t)\|_2 - \text{tr}\left(P_k(t)\right), \quad (31)$$

while eq. (7) becomes

$$\|W_k(t)\|_F^2 = \|W_{k+1}(t)\|_F^2 + \|W_k(0)\|_F^2 - \|W_{k+1}(0)\|_F^2 - \text{tr}\left(P_k(t)\right) + \text{tr}\left(Q_{k+1}(t)\right). \quad (32)$$

Summing eq. (31) and eq. (32) from $k$ to $L-1$, and invoke eq. (30), we get

$$\|W_k(t)\|_F^2 - \|W_k(t)\|_2^2 \leq D - \text{tr}\left(P_k(t)\right) + \text{tr}\left(Q_L(t)\right) + \sum_{k'=k}^{L-1} \text{tr}\left(P_{k'}(t)\right) \leq D + 2\mathcal{R}\left(W(0)\right).$$

To prove singular vectors get aligned, we can still proceed in nearly the same way as in the proof of Lemma 2. eq. (9) becomes

$$u_k^\top W_{k+1}^\top W_{k+1} u_k \geq \sigma_k^2 - \|W_k(0)\|_2^2 - \|Q_{k+1}(t)\|_2, \quad (33)$$

while eq. (10) becomes

$$u_k^\top W_{k+1}^\top W_{k+1} u_k \leq \langle u_k, v_{k+1} \rangle^2 \sigma_{k+1}^2 + D + 2\mathcal{R}\left(W(0)\right). \quad (34)$$

Combining eq. (33) and eq. (34)

$$\sigma_k^2 \leq \langle u_k, v_{k+1} \rangle^2 \sigma_{k+1}^2 + D + 2\mathcal{R}\left(W(0)\right) + \|Q_{k+1}(t)\|_2 + \|W_k(0)\|_2^2. \quad (35)$$

Similar to eq. (33), we can get

$$\sigma_k^2 \geq v_{k+1}^\top W_k W_k^\top v_{k+1} \geq \sigma_{k+1}^2 - \|W_{k+1}(0)\|_2^2 - \|P_k(t)\|_2,$$

and thus eq. (12) becomes

$$\frac{\sigma_k^2}{\sigma_{k+1}^2} \geq 1 - \frac{\|W_{k+1}(0)\|_2^2 + \|P_k(t)\|_2}{\sigma_{k+1}^2}. \quad (36)$$

Combining eq. (35) and eq. (36), we get

$$\langle u_k, v_{k+1} \rangle^2 \geq 1 - \frac{D + \|W_k(0)\|_2^2 + \|W_{k+1}(0)\|_2^2 + 3\mathcal{R}\left(W(0)\right)}{\sigma_{k+1}^2}.$$

The final claim of Lemma 8 can be proved in exactly the same way as Lemma 2. □

*Proof of Theorem 3.* Summing eq. (32), we know that for any two different layers $j > k$,

$$\|W_k(t)\|_F^2 - \|W_j(t)\|_F^2 = \|W_k(0)\|_F^2 - \|W_j(0)\|_F^2 - \text{tr}\left(P_k(t)\right) + \text{tr}\left(Q_j(t)\right).$$

Recall eq. (30), we know that

$$\left|\left(\|W_k(t)\|_F^2 - \|W_j(t)\|_F^2\right) - \left(\|W_k(0)\|_F^2 - \|W_j(0)\|_F^2\right)\right| \leq 2\mathcal{R}\left(W(0)\right). \tag{37}$$

In other words, the difference between the squares of Frobenius norms of any two layers is still bounded.

The proof then goes in the same way as the proof of Theorem 1. Suppose the risk is always above $\epsilon > 0$. Then there exists some $c(\epsilon) > 0$ such that $\|\nabla\mathcal{R}(w_{\text{prod}})\| \geq c(\epsilon)$. By Lemma 8, there exists some $C$ such that if $\min_{1 \leq k \leq L} \|W_k(t)\|_F > C$, $\|W_L(t) \cdots W_2(t)\| \geq C^L/2$. By eq. (37) and Lemma 7, $\sum_{t:\|W_k(t)\|_F \leq C \text{ for some } k} \eta_t$ is finite. On the other hand, by Lemma 7, $\sum_{i=0}^{\infty} \eta_t = \infty$, and thus $\sum_{t:\|W_k(t)\|_F > C \text{ for all } k} \eta_t = \infty$. Therefore we have, by invoking eq. (26),

$$\begin{aligned}
2\mathcal{R}\left(W(0)\right) &\geq \sum_{t=0}^{\infty} \eta_t \left\|\mathcal{R}\left(W(t)\right)\right\|^2 \\
&\geq \sum_{t=0}^{\infty} \eta_t \left\|\frac{\partial\mathcal{R}}{\partial W_1(t)}\right\|^2 \\
&\geq c(\epsilon)\frac{C^L}{2} \sum_{t:\|W_k(t)\|_F > C \text{ for all } k} \eta_t \\
&= \infty,
\end{aligned}$$

which is a contradiction. Therefore $\mathcal{R}\left(W(t)\right) \to 0$, and since it has no finite optimum, $\|W_k\|_F \to \infty$. The other results follow from Lemma 7. $\qquad\square$

*Proof of Theorem 4.* Recall that

$$\frac{\partial\mathcal{R}}{\partial W_1} = W_2^\top \cdots W_L^\top \nabla\mathcal{R}(w_{\text{prod}}),$$

and thus

$$\begin{aligned}
\|W_1(t+1)\|_F^2 &= \|W_1(t)\|_F^2 - 2\eta_t\left\langle W_1(t), \frac{\partial\mathcal{R}}{\partial W_1(t)}\right\rangle + \eta_t^2\left\|\frac{\partial\mathcal{R}}{\partial W_1(t)}\right\|_F^2 \\
&= \|W_1(t)\|_F^2 - 2\eta_t\left\langle w_{\text{prod}}(t), \nabla\mathcal{R}\left(w_{\text{prod}}(t)\right)\right\rangle + \eta_t^2\left\|\frac{\partial\mathcal{R}}{\partial W_1(t)}\right\|_F^2.
\end{aligned}$$

If $\langle w_{\text{prod}}, z_i\rangle \geq 0$ for all $i$, then $\|W_1(t+1)\|_F \geq \|W_1(t)\|_F$.

Also recall that $\Pi_\perp W_1(t)$ denote the projection of rows of $W_1(t)$ onto $\bar{u}^\perp$, the orthogonal complement of $\text{span}(\bar{u})$. We have

$$\begin{aligned}
\|\Pi_\perp W_1(t+1)\|_F^2 &\leq \|\Pi_\perp W_1(t)\|_F^2 - 2\eta_t\left\langle \Pi_\perp W_1(t), \frac{\partial\mathcal{R}}{\partial W_1(t)}\right\rangle + \eta_t^2\left\|\frac{\partial\mathcal{R}}{\partial W_1(t)}\right\|_F^2 \\
&= \|\Pi_\perp W_1(t)\|_F^2 - 2\eta_t\left\langle \Pi_\perp w_{\text{prod}}(t), \nabla\mathcal{R}\left(w_{\text{prod}}(t)\right)\right\rangle + \eta_t^2\left\|\frac{\partial\mathcal{R}}{\partial W_1(t)}\right\|_F^2.
\end{aligned} \tag{38}$$

Invoking eq. (26) again gives

$$\eta_t^2\left\|\frac{\partial\mathcal{R}}{\partial W_1(t)}\right\|_F^2 \leq \eta_t\left\|\nabla\mathcal{R}\left(W(t)\right)\right\|^2 \leq 2\left(\mathcal{R}\left(W(t)\right) - \mathcal{R}\left(W(t+1)\right)\right). \tag{39}$$

The proof then goes in almost the same way as the proof of Theorem 2. For any $\epsilon > 0$, we can find some large enough time $t_0$, such that for any $t \geq t_0$,

1. $\|\Pi_\perp W_1(t)\|_F/\|W_1(t)\|F \geq \epsilon$ implies that $\left\langle \Pi_\perp w_{\text{prod}}(t), \nabla\mathcal{R}\left(w_{\text{prod}}(t)\right)\right\rangle \geq 0$.

2. $\langle w_{\text{prod}}(t), z_i\rangle \geq 0$ for all $i$, and thus $\|W_1(t+1)\|_F \geq \|W_1(t)\|_F$.

3. $\|W_1(t)\|_F \geq {}^{1+\sqrt{2\mathcal{R}(W(0))}}/_\epsilon$.

Suppose at some time $t_1 \geq t_0$, $\|\Pi_\perp W_1(t_1)\|_F/\|W_1(t_1)\|_F \geq \epsilon$. As long as this still holds, in light of bullet (1) above, eq. (38) and eq. (39), $\|\Pi_\perp W_1\|_F^2$ will increase by at most $2\mathcal{R}\left(W(t_1)\right) \leq 2\mathcal{R}\left(W(0)\right)$. On the other hand, $\|W_1\|_F \to \infty$, and thus there exists some $t_2 > t_1$ such that $\|\Pi_\perp W_1(t_2)\|_F/\|W_1(t_2)\|F < \epsilon$.

Let $t_3$ denote the smallest time after $t_2$ such that $\|\Pi_\perp W_1(t_3)\|_F/\|W_1(t_3)\|_F \geq \epsilon$ (if it exists). Recall that $\|W_1(t+1)\|_F \leq \|W_1(t)\|_F + 1$ for any $t \geq 0$, and $\|W_1(t+1)\|_F \geq \|W_1(t)\|_F$ for any $t \geq t_0$, we have

$$\frac{\|\Pi_\perp W_1(t_3)\|_F}{\|W_1(t_3)\|_F} \leq \frac{\|\Pi_\perp W_1(t_3)\|_F}{\|W_1(t_3-1)\|_F} \leq \frac{\|\Pi_\perp W_1(t_3-1)\|_F + 1}{\|W_1(t_3-1)\|_F} < \epsilon + \frac{1}{\|W_1(t_3-1)\|_F}.$$

After $t_3$, $\|\Pi_\perp W_1\|_F^2$ will increase by at most $2\mathcal{R}\left(W(0)\right)$, and thus $\|\Pi_\perp W_1\|_F$ will increase by at most $\sqrt{2\mathcal{R}\left(W(0)\right)}$. Therefore, for any $t_4 \geq t_3$, as long as $\|\Pi_\perp W_1(t_4)\|_F/\|W_1(t_4)\|_F \geq \epsilon$, we have

$$\frac{\|\Pi_\perp W_1(t_4)\|_F}{\|W_1(t_4)\|_F} \leq \frac{\|\Pi_\perp W_1(t_4)\|_F}{\|W_1(t_3)\|_F}$$

$$\leq \frac{\|\Pi_\perp W_1(t_3)\|_F + \sqrt{2\mathcal{R}\left(W(0)\right)}}{\|W_1(t_3)\|_F}$$

$$\leq \epsilon + \frac{1}{\|W_1(t_3-1)\|_F} + \frac{\sqrt{2\mathcal{R}\left(W(0)\right)}}{\|W_1(t_3)\|_F} \leq 2\epsilon,$$

since $\|W_1(t)\|_F \geq {}^{1+\sqrt{2\mathcal{R}(W(0))}}/_\epsilon$ after $t_0$. In other words,

$$\limsup_{t\to\infty} \frac{\|\Pi_\perp W_1\|_F}{\|W_1\|_F} \leq 2\epsilon.$$

Since $\epsilon$ is arbitrary, we have

$$\limsup_{t\to\infty} \frac{\|\Pi_\perp W_1\|_F}{\|W_1\|_F} = 0,$$

and thus $\lim_{t\to\infty}\left|\langle v_1, \bar{u}\rangle\right| = 1$.

$\square$

*Proof of Corollary 2.* The proof is analogous to that of Corollary 1, except using Theorem 4 in place of Theorem 2. $\square$

