# OpenReview forum: "Gradient descent aligns the layers of deep linear networks"
_ICLR.cc/2019/Conference_

### Official Review · AnonReviewer2 · 2018-10-29

**Rating:** 6
**Confidence:** 5

**Review:**

Summary:
This paper studies the properties of applying gradient flow and gradient descent to deep linear networks on linearly separable data. For strictly decreasing loss like the logistic loss, this paper shows 1) the loss goes to 0, 2) for every layer the normalized weight matrix converges to a rank-1 matrix 3) these rank-1 matrices are aligned. For the logistic loss, this paper further shows the linear function is the maximum margin solution.

Comments:
This paper discovers some interesting properties of deep linear networks, namely asymptotic rank-1, and the adjacent matrix alignment effect. These discoveries are very interesting and may be useful to guide future findings for deep non-linear networks. The analysis relies on many previous results in Du et al. 2018, Arora et al. 2018 and Soudry et al. 2017  authors did a good job in combing them and developed some techniques to give very interesting results.
There are two weaknesses. First, there is no convergence rate. Second, the step size assumption (Assumption 5) is unnatural. If the step size is set proportional to 1/t or 1/t^2  does this setup satisfies this assumption?

Overall I think there are some interesting findings for deep linear networks and some new analysis presented, so I think this paper is above the bar.
However, I don't think this is a strong theory people due to the two weakness I mentioned.

---

> ### Author Response · Authors · 2018-11-13
> **Response to AnonReviewer2**
>
> We thank the reviewer for their time and careful comments.
>
> We disagree that "all the analyses have appeared in previous papers".  We wish to communicate with the reviewer during this feedback phase in order to come to a consensus on this comment, and subsequently update the submission to accurately present what is new and what is old in the analysis.
>
> To start this discussion, we clarify how our analysis goes beyond what was known.
> (1) We first argue that Theorem 1 (and analogously Theorem 3) sharply depart from prior work.  In particular, the tools from (Arora et al., 2018; Du et al., 2018) are only used at the beginning of Lemma 2, and moreover Lemma 2 is not nearly strong enough to prove Theorem 1: first, it is still possible for the iterates to get trapped in saddle points, or more generally in a bounded domain; second, even if the iterates grow unboundedly, the risk may still not converge to zero. These problems are handled in the proofs of Lemma 1 and Theorem 1 respectively, using techniques which have not previously appeared.
> (2) Theorem 2 (and Theorem 4) also depart from prior work.  We invoke a lemma of Soudry et al. (2017) in our technical Lemma 3; otherwise, the proofs of Lemma 4 and Theorem 2 are new.  Indeed, the work of Soudry et al. (2017) is for linear predictors, whereas we consider deep linear networks.
>
> On a separate note, we agree that producing rates and practical step sizes would be ideal. However, the analysis of gradient flow is already an interesting stepping stone, indeed one which is the main topic of prior work (Arora et al., 2018; Du et al., 2018).  Our step sizes are not standard, but we note that they can be computed easily via the expression for beta(R) given in Lemma 5.
>
> We thank the reviewer once again, and look forward to further comments!

---

> > ### Comment · AnonReviewer2 · 2018-11-22
> > **Thanks for your response**
> >
> > Thanks for your response. I have changed my review and I think your paper is above the bar.

---

> > > ### Author Response · Authors · 2018-11-26
> > > **Response to AnonReviewer2**
> > >
> > > We thank the reviewer for their comments. As detailed in our "common response", we have updated our "Related Work" and "Summary and Future Directions" in response to your comments.  Thank you for your time!

---

### Official Review · AnonReviewer1 · 2018-11-02
**Strong guarantees for deep linear networks on separable data**

**Rating:** 9
**Confidence:** 4

**Review:**

In this work the authors prove several claims regarding the inductive bias of gradient descent and gradient flow trained on deep linear networks with linearly separable data. They show that asymptotically gradient descent minimizes the risk, each weight matrix converges to its rank one approximation and the top singular vectors of two adjacent weight matrices align. Furthermore, for the logistic and exponential loss the induced linear predictor converges to the max margin solution.

This work is very interesting and novel. It provides a comprehensive and exact characterization of the dynamics of gradient descent for linear networks. Such strong guarantees are essential for understanding neural networks and extremely rare in the realm of non-convex optimization results. The work is a major contribution over the paper of Gunasekar et al. (2018) which assume that the risk is minimized. The proof techniques are interesting and I believe that they will be useful in analyzing neural networks in other settings.

Regarding Lemma 3, the proof is not clear. Lemma 8 does not exist in the paper of Soudry et al. (2017). It is also claimed that with probability 1 there are at most d support vectors. How does this relate with assumption 3, which implies that there are at least d support vectors?

-------Revision---------

Thank you for the response. I have not changed the original review.

---

> ### Author Response · Authors · 2018-11-13
> **Response to AnonReviewer1**
>
> We thank the reviewer for their time and careful comments.
>
> The correct reference for "Lemma 8" of Soudry et al. (2017) is either Lemma 8 of their ICLR submission ( https://openreview.net/forum?id=r1q7n9gAb ), alternatively Lemma 12 in their current (as of March 18) arxiv version ( https://arxiv.org/abs/1710.10345v3 ). We do not require the full strength of this lemma; we only need all support vectors to have positive dual variables with probability 1. While this lemma is a property of support vectors, our Assumption 3 is on the relation between support vectors and nonsupport vectors; we do not necessarily need the support vectors to span the whole space, it is enough if they span the same space as the data, even if this is a subspace of dimension smaller than the ambient dimension.
>
> We thank the reviewer for their support.  We believe that our techniques will be helpful in understanding nonlinear networks, and that alignment results there will help with other problems, for instance generalization.
>
> We will be following and responding to comments throughout this feedback phase, and welcome all further comments from the reviewer!

---

### Official Review · AnonReviewer3 · 2018-11-03
**An insightful result**

**Rating:** 7
**Confidence:** 4

**Review:**

This paper analyzes the asymptotic convergence of GD for training deep linear network for classification using smooth monotone loss functions (e.g., the logistic loss). It is not a breakthrough, but indeed provides some useful insights.

Some assumptions are very restricted: (1) Linear Activation; (2) Separable data. However, to the best of our knowledge, these are some necessary simplifications, given current technical limit and significant lack of theoretical understanding of neural networks.

The contribution of this paper contains multiple manifolds: For Deep Linear Network, GD tends to reduce the complexity:
(1)	Converge to Maximum Margin Solution;
(2)	Tends to yield extremely simple models, even for every single weight matrix.
(3)	Well aligned means handle the redundancy.
(4)	Experimental results justify the implication of the proposed theory.

The authors use gradient flow analysis to provide intuition, but also present a discrete time analysis.

The only other drawbacks I could find are (1) The paper only analyze the asymptotic convergence; (2) The step size for discrete time analysis is a bit artificial. Given the difficulty of the problem, both are acceptable to me.

---

> ### Author Response · Authors · 2018-11-13
> **Response to AnonReviewer3**
>
> We thank the reviewer for their time and careful comments.
>
> We agree with the reviewer's criticisms.  We hope to work with nonlinear networks, practical step sizes, and provide rates in follow-up work.
>
> We thank the reviewer for their support.  As we mentioned to AnonReviewer1, we believe these tools can also help in the analysis of nonlinear networks, and these alignment results can then be used to derive refined generalization bounds.
>
> We thank the reviewer once again, and invite them to provide further comments during this feedback period!

---

### Author Response · Authors · 2018-11-26
**Overview of our Revision**

We have uploaded a minor revision of our paper:
- We have adjusted our concluding "Summary and Future Directions" section to highlight the need for convergence rates and practical step sizes.
- Due to a discussion with AnonReviewer2, we have foreshadowed our use of prior work in our "Related Work" subsection.
- Due to a discussion with AnonReviewer1, we have included the exact version number of (Soudry et al., 2017), and now refer to their "Lemma 12" rather than "Lemma 8".
- We have fixed a few typos.

We thank the reviewers for their time and feedback!

---

### Meta-Review · Area_Chair1 · 2018-12-11
**ICLR 2019 decision**

**Confidence:** 4
**Recommendation:** Accept (Poster)

**Metareview:**

This paper studies the behavior of weight parameters for linear networks when trained on separable data with strictly decreasing loss functions. For this setting the paper shows that the gradient descent solution converges to max margin solution and each layer converges to a rank 1 matrix with consequent layers aligned.  All reviewers agree that the paper provides novel results for understanding implicit regularization effects of gradient descent for linear networks. Despite the limitations of this paper such as studying networks with linear activation, studying gradient descent not with practical step sizes, assuming data is linearly separable, reviewers find the results useful and a good addition to existing literature.